# Conceptual Framework of a Psychotherapeutic Consultation in the Workplace: A Qualitative Study

**DOI:** 10.3390/ijerph192214894

**Published:** 2022-11-12

**Authors:** Eva Rothermund, Tim Pößnecker, Andreas Antes, Reinhold Kilian, Franziska Kessemeier, Jörn von Wietersheim, Dorothea Mayer, Monika A. Rieger, Harald Gündel, Michael Hölzer, Elisabeth M. Balint, Kathrin Mörtl

**Affiliations:** 1Department of Psychosomatic Medicine and Psychotherapy, Ulm University Medical Center, Albert-Einstein-Allee 23, 89081 Ulm, Germany; 2Leadership Personality Center Ulm (LPCU), Ulm University, Kornhausgasse 9, 89073 Ulm, Germany; 3Department of Psychiatry II, Ulm University Medical Center at BKH Günzburg, Lindenallee 2, 89312 Günzburg, Germany; 4Health and Safety Sindelfingen, Daimler AG, Bela-Barenyi-Straße, 71059 Sindelfingen, Germany; 5Institute for Occupational and Social Medicine and Health Services Research, University Clinic Tübingen, Wilhelmstraße 27, 72074 Tübingen, Germany; 6Sonnenbergklinik, ZfP Südwürttemberg, Christian-Belser-Straße 79, 70597 Stuttgart, Germany; 7Faculty of Psychotherapy Science, Sigmund Freud University, Freudplatz 1, 1020 Vienna, Austria

**Keywords:** company health promotion, complex intervention, depression, early intervention, grounded theory methodology, health services research, mental health

## Abstract

(1) Background: A new health care offer called ‘psychotherapeutic consultation in the workplace’ is an early and effective intervention for employees with common mental disorders. Although cost-effective, it lacks a broader roll-out. This might be attributable to undefined context, mechanisms of change, and a lack of communication; therefore, this study explores how the new model works and where problems occur. (2) Methods: Semi-structured interviews on motivation, expectations and experiences with 20 involved experts. Experts were members of the company health promotion team, service users, and cooperating mental health specialists. Analysis was conducted with ATLAS.ti. (3) Results: The conceptual framework comprises three main topics: (a) structured implementation concept; (b) persons involved, shaping the concept’s processes; (c) and meaning and function of the offer within the given context. Concerning (c) we found three potential areas of conflict: (1) intra-corporate conflicts, (2) conflicts between company and employee, (3) and conflicts between the company health promotion and the health care system. Category (c) comprises the offer’s core characteristics which were described as low-threshold and preventive. Furthermore, the offer was perceived as convenient in handling, confidential, and having immediate impact on a person’s well-being. (4) Conclusions: Here we define structures, address the needs of the involved persons, and communicate foreseeable areas of conflict influences whether the implementation of the intervention succeeds.

## 1. Introduction

Common mental disorders (CMDs) such as depression and anxiety place a substantial burden on the economies of many developed countries [1]. Despite the availability of therapies in high income countries, only 30% of persons affected by a major depressive disorder are, for example, in treatment in Germany [2]. Barriers to utilization are the fear of stigmatization [3,4,5] and gender role expectations [6,7,8]. The resulting occupational dysfunction of untreated illness may lead to poverty, social isolation, and enhancement of mental symptoms [9,10].

Psychotherapeutic consultation in the workplace (PSIW) stands for mental health care that is offered at the worksite. In the context of this collaborative care model, a mental health specialist visits a company to offer consultation for employees. This kind of model resembles the consultation–liaison model applied in psychosomatic medicine [11], the outreach work in social work [12], the employee assistance programs (EAPs) [13] or the workplace counselling [14,15]. PSIW is provided by a psychological or medical mental health care specialist. The work of this specialist, although not employed by the host company, may be remunerated by the company (as an external contractor) or covered by the company’s health insurance fund. Informed consent provided, the mental health specialist is allowed to exchange information with the company physician about the service user (e.g., working conditions, clinical data, and organizational options about how to change the working position or reduce working hours for a certain time). This service is free of charge for all members of staff. Staff members are usually informed about the service by the company physician. Whereas self-referral is possible in some companies, in others employees must be referred to PSIW by the company physician. Wherever self-referral is possible, information about PSIW (type of service, location, and appointment-making) is provided online or on paper. The consultation comprises 1–2 sessions, each 50–60 min. The main difference between PSIW and treatment as usual which is psychotherapeutic outpatient care (PSOC) is the setting and location of the offer and a focus on work-related problems in PSIW. While PSIW is worksite-based within the facilities of the company physician, PSOC takes place in the outpatient department of the cooperating hospital. The content of the intervention results from the given structure (brief intervention, location, and context), it represents an extended version of the ‘initial psychiatric interview’ [16] and is therefore an approach independent of various therapy schools. In contrast to PSOC, PSIW puts a special emphasis on vocational issues; however, both are based on the four following psychotherapeutic core elements: 1. Relationship development between employee and health care specialist; 2. diagnostic assessment including work-related factors and work functioning; 3. promotion of self-management strategies being supportive and resource-oriented (i.e., by implementing an individual explanatory model, which takes work-related stressors into account); and 4. motivation and support for further steps (i.e., treatment or making the health care system accessible at all in helping to find the right contact) [17].

Collaborative worksite mental health care allows to improve mental health, facilitates return to work [14,15,18,19,20,21], or reduces recurrent sickness absence due to CMD [22]. Recently, we proved that the distinct PSIW model [23] enhances early intervention [24,25] for employees not yet sick-listed while being as effective as PSOC, a regional comparable offer in regular care [26].

Although being efficient and cost-effective [27], such models lack a broader roll out [28,29] due to the fact that the PSIW model as a pilot solution has so far only been reported in the scientific field. Thus, a reliable overall concept needs to be elaborated and made accessible to companies. Second, so far there is not a defined model of reimbursement by the German Social Code. The gap of systematic implementation might result from the large variety of different models. They comprise various collaborations [30] and focus on different stages of treatment, rarely on secondary prevention but mostly on tertiary prevention [31]. Furthermore, the models can only be found as pilot studies in the research field [23,32,33,34,35] rendering the access difficult to companies.

In the study context of the German health care system, patients with common mental disorders are regularly treated with psychotherapy by physicians specializing in psychiatry or psychosomatic medicine or by psychological psychotherapists. Treatment is usually delivered through private practices, the outpatient clinics of psychosomatic hospitals, and psychosomatic departments or psychosomatic outpatient clinics at general hospitals [36]. In the German health care system PSOC is covered by statutory and private health insurance and nearly 100% of the population is covered by health insurance.

When implementing a new health care offer within an organization, the challenge consists in understanding the complex field being characterized by an intensive interplay of medical, psychological, social, and cultural factors [28]. A few PSIW-like models have been described in scientific literature for Germany starting with reporting the local history of implementation at a large automobile manufacturer, specific work-related topics/issues at that site [37], and necessities-integrated care models at the intersectoral interface such as transparency and confidentiality in communication [37,38]. The opportunities and limitations of the services offered were firstly analyzed in a qualitative analysis by Preiser and colleagues [33,39]. PSIW was also influenced by evidence from international care offers combining symptom and work-focused interventions [21] describing a model with occupational physicians trained in diagnosis and treatment of mental health problems in the Netherlands. Another Dutch model [20] reported a collaborative care model which combined sickness guidance by occupational physicians with 6–12 sessions of problem-solving treatment, manualized self-help, workplace intervention, and antidepressant medication. A work-directed and workplace-based model in the USA was reported from Lerner [40] focusing on work participation and the reduction in on-work limitation. Interventions strongly involving the workplace with tailored interventions in the return to work process after sick leave due to mental health issues were described for Denmark [41] and Norway [42]. Besides Preiser [33,39], none of these studies have described in-depth the model and its implementation so far even though it is growing such as our own model [43], and is subject to randomized controlled investigations [42,44].

Due to a missing conceptual framework, the application or further implementation of an intervention such as PSIW is largely limited [45]. As a result, each company must find its own solution and unique evaluation concept [14]. To explore how and why the intervention works, the process of implementation, the settings, the characteristics of the individuals involved, and the intervention characteristics need to be defined [46]. No comprehensive description of how and why any of the above-mentioned worksite offers work has been published so far [1,23,32,33,34,35]; however, the issue of whether services should better be based within the organization or externally has been debated [39]. In addition, the need for further investigation has been identified [15]. In order to understand and bridge the research–practice gap, we started dialogue with PSIW service users, stakeholders within the company organizing the offer (e.g., company doctors, supervisors, or social workers), and external psychotherapists.

## 2. Materials and Methods

### 2.1. Design and Setting

As design we chose an explorative, cross-sectional, qualitative case study. It was part of the mixed methods study entitled ‘Psychosomatic consultation in the workplace—a new model of care at the interface of company supported mental health care and consultation-liaison psychosomatic’ [38]. For the international reader, we decided to re-label the intervention into ‘psychotherapeutic consultation’ instead of ‘psychosomatic consultation’ since the main procedure has a psychotherapeutic character which can be widely understood.

Three companies piloting the PSIW model and collaborating with external mental health specialists were included in the study. The health care service has been established regardless from study conditions and is based on individual contracts between local health providers and the company or the company’s health insurance fund.

### 2.2. Participants and Data Collection

#### 2.2.1. Participants and Location of Study

Any person involved in the new model of care was qualified as an expert. We conducted the study with persons from three companies and two hospitals in two regions in southern Germany. The participating companies and hospitals, representing two local networks of collaboration decided to start a research project funded by a third party [47]. The company stakeholders were either employed by an automobile manufacturer, a metal works company, or a company developing security systems.

At that time, the investigated company sites employed between 80 and 90% men working in technique-based occupations (e.g., engineers or blue-collar workers). Representatives of the company’s health promotion team were recruited in all three companies. External experts in mental health were working in two clinics and collaborating in three companies. Service users were recruited consecutively from one company specializing in security systems. Workplace health promotion representatives are persons within the company who foster and implement interventions or policies within the organization to maintain and promote employees’ health. Within this framework, the teams were comprised of occupational health physicians, members of the work council, and HR professionals. Table 1 provides an overview of the sample characteristics. Possibly due to the age structure of patients treated in the PSIW [24] and professionals working there, which are both of older than in regular PSOC, the age category of 19–30 years was not present in our sample.

#### 2.2.2. Interviewers

The two participating outpatient clinics provide mental health care specialists from the Department of Psychosomatic Medicine and Psychotherapy at Ulm University Medical Center and Sonnenberg Klinik, Division of Psychosomatic Medicine of the ZfP Südwürttemberg, Stuttgart, a local provider for in- and outpatient mental health care.

At that time, the investigated company sites employed between 80 and 90% men working in technique-based occupations (e.g., engineers or blue-collar workers). Representatives of the company’s health promotion team were recruited in all three companies. External experts in mental health were working in two clinics and collaborating in three companies. Service users were recruited consecutively from one company specializing in security systems. Workplace health promotion representatives are persons within the company who foster and implement interventions or policies within the organization to maintain and promote employees’ health. Within this framework, the teams were comprised of occupational health physicians, members of the work council and HR professionals.

#### 2.2.3. Data Collection

The goal was to conduct 20 semi-structured interviews (see Table 1). While conducting the interviews it seemed as if theoretical saturation was reached after conducting the planned number of interviews, since no additional relevant categories emerged from our data anymore.

On average, an interview took 24 min ranging from 13 to 38 min. Interviews were conducted by AA at the Department of Psychosomatic Medicine and Psychotherapy in Ulm or at the interviewees’ workplaces from November 2012 until February 2014. The PSIW offer was implemented in each company at least 1 year before (i.e., 2010) we started with the interviews. The quantitative data assessment took place from November 2011 until June 2013.

#### 2.2.4. Interview Guideline

At the beginning, a semi-structured interview guide (see Appendix A) was developed regarding expectations, experiences, and motivation of experts taking part in the interview. The key questions were tailored to each single group but all covered the same topics. The following questions were asked: 1. Why did you participate in the offer (motivation)? 2. What did you expect? 3. How did you experience the offer? 4. How would you rate your overall satisfaction with this experience?

Before establishing the interview guide for the rest of the process, adjustments were made after the first interviews. During the process, data were obtained via interviews and afterwards digitally audio recorded, anonymized, and transcribed verbatim by a professional transcription service. The transcripts were imported into ATLAS.ti for further analysis.

### 2.3. Data Analysis

The basic procedural rules of grounded theory [48] were applied in accordance with the application description of grounded theory method by Dourdouma and Mörtl [49]. The processes of analysis and data collection were synchronized. In the open coding process, the data were read carefully to identify text units that were meaningful regarding the research topic. The resulting codes reflected the original statement as close as possible (paraphrasing). On the basis of these paraphrases, first codes were formed. In the course of the analysis the resulting coding system was constantly revised. Codes with similar content were summarized to higher level categories. These categories were then structured in an axial coding process and further integrated. If further paraphrases occurred in the course of analysis, they were added to the coding scheme. At the beginning of the analysis, transcripts were coded simultaneously by two researchers (AA and ER). Throughout the process, the obtained coding systems were compared and the allocation of meaning units to particular categories was discussed in an interdisciplinary team with different backgrounds. In the next step, the core category was defined. The core category stands for the most central code summarizing and heading the main theme or fundamental principle.

### 2.4. Researcher Reflexivity

Reflexivity was maintained by the research team during the process of analyzing and challenging established assumptions. The study design was developed by ER, RK, HG, MH, and DM. ER, TP, EB, HG, JW, and MH were involved in organizing PSIW services. DM accompanied this process as an occupational health physician. In addition, DM, KM, and AA, who were not involved in the offer, took care of the data analysis. A group of various experts in sociology, psychology, psychosomatic medicine, and occupational medicine developed the concept in cooperation.

## 3. Results

The conceptual framework comprises three main elements that are important in implementing an intersectoral offer of cooperation between statutory health care and company health promotion (Figure 1). Firstly, there is the need for a framework which structures the implementation process (column A). Secondly, there are persons involved, in this context also called persons involved (column B), who induce and shape the processes within the implemented framework. These persons involved interact with each other in their function as a service user, stakeholder within the company, or external psychotherapist. Furthermore, their interaction is led by their perceptions. Thirdly, the meaning and function of the offer is reflected by the intellectual framework that is influenced by the societal context and how mental health conditions are perceived in society at large but also at the workplace (column C). The different situations, requirements, and interests in between which the offer is situated create various areas of potential conflict. They define important relational contacts that emerge from the interaction but influence them as well. PSIW was described as being low-threshold and preventive. It was perceived as convenient and simple in handling and stated to be confidential. It is supposed to work as a quick fix and is perceived as having direct and immediate impact on a person’s well-being. The corresponding quotes are displayed in Appendix B, Table A1, Table A2 and Table A3.

### 3.1. Implementation (A1 and A2)

Implementation comprises the steps that should be performed in preparation for the process from an interviewee’s perspective and which, from an organizational and structural point of view, have proved to be helpful in the process of implementation.

#### 3.1.1. Preparation (A1)

At the beginning of the implementation process a needs assessment is required and distinct key events need to be defined (Table A1, Quote A1.1). An overall rising but unmet need for adequate mental health care for common mental disorders was stated. Some interviewees drew parallels to the public discussion of rising numbers of common mental disorders, others remembered real cases they had dealt with. Critical voices discussed that offering mental health care at the workplace itself promotes a rising need for treatment in the sense of medicalization (Table A1, Quote A1.1). In order to determine mental health at work, a risk assessment comprising psychosocial health and health-related quality of life measures as well as psychosocial work-stressors was made to demonstrate the need for mental health offers within company health promotion. Trainings for managers in order to sensitize them for persons in need, or to witness somebody in need for mental health care proved to be positive for the implementation (Table A1, Quote A1.2). PSIW has been described as one component of the company mental health promotion along with primary preventive offers, manager trainings, etc. An existing company health promotion program was found to be a good basis to obtain the offer started considering the fact of company members already being aware of the importance of implementing health promotion offers (Table A1, Quote A1.3). To merely prepare the ground for PSIW, initiatives within the company but also from external stakeholders such as mental health professionals (holding, for example, lectures on mental health) were considered to be helpful with regard to enhanced knowledge about mental health literacy within the company (Table A1, Quote A1.4).

#### 3.1.2. Organizational and Structural Elements (A2)

Organizational and structural elements were reported as the framework for PSIW. Eight essential issues were raised and should be defined in advance to set up the framework (Figure 1, Column A, A2): therapeutic dose, content and structure of the intervention, persons involved, place of offer (location), economic considerations, promotion of offer, pathway to care for employees, and company´s preconditions.

Content and structure were reported to follow the rationale of a diagnostic assessment. Basic knowledge about how mental stress can impact body functions was recommended to be transferred to the service user. In certain cases, it was recommended to impart basic knowledge about mental disorders and about treatment options to the service user. In order to motivate patients for further treatment, creating a positive treatment experience was considered to be an important part of the intervention (Quote A2.2). The question was raised whether PSIW should be situated on the company premises or outside of the company. Concerns about privacy on the one hand and convenience on the other hand were noted. The fear of being stigmatized by peers or supervisors in case of being seen when using this mental health care service was reflected by the interviewees. Keeping private matters private was indicated to be more difficult with PSIW being situated at the company site (Table A1, Quote A2.4). PSIW was reported as a pilot solution. Models involved in this study were reported to be covered mainly by the budget of the company or by the corresponding company health insurance fund. These investments were stated to be economically viable (Table A1, Quote A2.5). Events took place in the company in order to introduce the company health promotion representatives or even the corresponding external psychotherapists to all employees. Depending on the business culture, i.e., engineers in development initiating things by themselves or employees in production relying more on (health care) professionals, the pathway, i.e., entry to care was either performed by self-referral or due to the recommendation by health care staff. Medical referrals were initiated, in particular, by company doctors but also by persons working at the social service or by representatives for persons with disabilities (Table A1, Quote A2.7). Preconditions to be fulfilled by the company were facilities and appointment making. Furthermore, factory tours were testified to help the external psychotherapists to become more familiar with the working circumstances affecting the employees (Table A1, A2.8). For an overview of quotations on the above-mentioned points, see Appendix B, Table A1.

### 3.2. Persons Involved and Perceptions (B)

Reflections on the role of the occupational health service, the company, the service users, and the external mental health specialist are outlined together with general statements (Figure 1 Column B with corresponding Quotes in Table A2).

#### 3.2.1. Occupational Health Service (B1)

The company doctor was perceived as the primary in-house contact for all health issues by the company health promotion team and the external psychotherapists. Service users did not share this view (see Table A2). In the vocational setting, this role appeared comparable to a primary care physician’s role outside the company. The company doctor is supposed to be an easily accessible person within the company with whom a wide variety of issues can be discussed. A relationship of trust, for example, was reported to often develop by means of regular routine check-ups (Table A2, Quote B1.1).

The occupational health service was considered to pave the way for PSIW. In detail, the utilization of PSIW was supported by structures such as the occupational health service facilities, procedures such as appointment making and encouraged by the atmosphere of confidentiality and preparatory consultation by the occupational health physician (Table A2, Quote B1.3).

#### 3.2.2. Company (B2)

The company’s motivation was subsumed under the heading ‘claim and benefit’: PSIW is expected to be a tool to tackle mental health problems at an early stage, to reduce presentism and sickness absence and thus, to promote the company’s economic benefit (Table A2, Quote B2.1). The risk of being a caretaker (Table A2, code ‘nursing framework’, B2.3) and a controller at the same time (Table A2, code ‘controlling framework’, B2.2) was discussed. The interviews (only from the company health promotion team and the external psychotherapists) revealed that mental ill-health is often not considered to be a genuine part of the company health promotion issues. The company health promotion representatives expressed that they sometimes felt unfamiliar dealing with employees with mental health issues and remembered situations of feeling helpless (Table A2, Quote B2.4).

#### 3.2.3. Service Users (B3)

According to the external psychotherapists and the company health promotion team, the characteristic service user was perceived as a certain type of employee who is unfamiliar with psychosomatic or mental health treatment. They appeared to need more time until they were ready to use such an offer, or rather, to talk openly about personal problems. Before seeking external advice, they pondered over the issue by themselves trying to solve it. PSIW was perceived to attract first time users for mental health care offers as well as to encourage and support persons who have already gained experience with the mental health care system (Table A2, Quote B3.1).

The nature of problems arising in PSIW was reported to often cover both areas, private and work-related issues. The level of severity was estimated to range from persons with mild problems to severe problems that can be classified as disorders and chronically recurrent conditions. External psychotherapists expressed their astonishment at meeting individuals in PSIW in quite severe conditions who were still at work. According to the experiences reported, work-related problems often motivated people to take the last step forward using the offered service (Quote B3.2).

Service users felt empowered by the PSIW offer (Table A2, Quote B3.3). Therapists considered work-related issues as very important in therapy. Vice versa, issues that were brought up in PSIW were considered to have more chances to be taken seriously by other company representatives (Table A2, Quote B3.5). External psychotherapists stated that service users should fully benefit with regard to regaining or preserving their mental health and work ability instead of being reported ill and therefore placing a burden on their colleagues (Table A2, Quote B3.4). However, this topic was not brought up by the service users.

#### 3.2.4. External Mental Health Specialist (B4)

The external mental health specialist´s role (for better readability hereinafter called ‘external psychotherapist’) was stated to require ongoing or acquired specialization in psychosomatic medicine (or a referring discipline such as psychotherapy or psychiatry). A clear definition of the role of an external psychotherapist has been requested by the service users and the members of the company health promotion team. The external psychotherapist should be the same person for a while and not being employed by the company to keep the distance (Quote B4.1). Furthermore, their role should imply knowing the company’s workflows and being informed about work-related mental health problems (Quote B4.2).

#### 3.2.5. General Perceptions of PSIW (B5)

Concerns and positive appraisals about the offer (Figure 1, Column B5) were stated. The following negative statements concerning PSIW were provided: experienced limited time frame, expected organizational challenges (e.g., cooperating external mental health specialists), unsuitable facilities, and disappointment (e.g., some employees expected more comprehensive support). After having used the offer, the limited power to influence the organizational part of work-related problems was criticized and some were afraid that PSIW can be misused to stay away from work. However, in total, many quotes proved the offer to be generally well-received. This is also reflected by the statements describing PSIW as sensible, appealing, very good, and an excellent thing (B5.3). For an overview of quotations on the above-mentioned points, see Appendix B, Table A2.

### 3.3. Meaning, Function and Context (C)

The complex of themes called ‘meaning, function, and context’ (Figure 1, Column C) is based on the structures and persons that are needed for PSIW. Opportunities and limitations of three potential areas of conflict (C3) that emerge between different poles are described. Moreover, the societal context (C2) in which the offer takes place is illustrated. Finally, the characteristics of PSIW (C3) that emerged as key features are defined (Table A3).

#### 3.3.1. Areas of Potential Conflict (C1)

##### C1.1 Priorities within the Company

Within the existing company health program, the offer was either perceived as competing or enriching. PSIW was described as an additional offer in an already existing corporate health promotion program. Before the new offer was introduced, other stakeholders took care of mental health issues. Company mental health promotion representatives as well as external psychotherapists expressed their concern about PSIW turning out to be a competing offer (Table A3, Quote C1.1_1, C1.1_2). On the other hand, PSIW was considered to complete the existing offers by adding components which so far had not been offered in this special field. This perception was expressed by interviewees from all three perspectives. The new and existing services were, contrary to the above-mentioned statements, reported to collaborate closely (Table A3, Quote C1.1_3, C1.1_4).

##### C1.2 Privacy and Support

Another area of potential conflict was perceived between the conflicting interests of privacy and support in close company proximity. Service users expressed their worries about detrimental effects of close company proximity. They were afraid that the company might ask for information about work-related problems via PSIW, which can result in disadvantages for the employee, or that an external psychotherapist might overreact in the unfamiliar setting of an enterprise (Table A3, Quote C1.2_1, C1.2_2). The benefits of close company proximity were mainly reported by external psychotherapists. Being there and having more insight, the psychotherapist can understand some problems better, integrate worksite issues into the consultation, and provided the patient consents, feedback on organizational issues (Table A3, Quote C1.2_3, C1.2_4).

##### C1.3 Company Health Promotion and Regular Care

Conflicting interests in this third area also emerged from the material. The external psychotherapists perceived PSIW as an intermediate stage in a stepped care approach. It was assumed to ease help-seeking behavior in the regular care system via mediating contacts, supporting appointment making, or informing about psychiatric or psychotherapeutic procedures to reduce fear. PSIW was expected to bridge that gap in case of longer waiting times for treatment in the secondary treatment system (Quote C1.3_1, Quote C1.3_2).

The topics were perceived to be the same in PSIW and regular care, while the changed structural preconditions in PSIW were reported to have an impact on how therapy was delivered: For example, seeing a patient less often or only in a very fixed time frame (Quote C1.3_3, Quote C1.3_4).

Offers in regular care were mentioned to be insufficient. The company mental health promotion team perceived the existing mental health care system as fear-inducing, and difficult to access and utilize. Inflexibility to use tailored interventions was reported by the external psychotherapists. Efforts to overcome this barrier were only experienced in cases with individuals with high levels of suffering. Bad experiences, stated by service users, enforced this impression (Quote C1.3_7 to Quote C1.3_9)

Too long waiting times in external offers were raised as a topic from different perspectives. It was noted that especially persons in need, with an ambivalent motivation for treatment, often gave up seeking help in the face of long wait times (Quote C1.3_10 and C1.3_11).

#### 3.3.2. Societal Context (C2)

The societal context is comprised of statements dealing with the work environment, stigmatization, and perception of mental health problems.

High workload and flexibility requirements for employees were stated as a challenging working environment and as a risk factor for developing mental health problems. The meritocratic labor market was blamed for leaving no space for slowdown or weakness. There was general agreement that more and more stress is put on the individual person these days. Service users perceived stress-related disorders as consequences resulting from these circumstances (Quote C2.1 and C2.2).

The stigma of mental health problems poses a barrier to qualify certain problems as mental health issues. Thus, it hinders help-seeking behavior for adequate treatment. The fear of jeopardizing their career was expressed. A different perception of PSIW is expected to change attitudes (Quote C2.3 and C2.4).

#### 3.3.3. Key Features of PSIW (C3)

With regard to the implemented structures, the interacting parties and the context workplace, key features of PSIW can be defined as follows:

A low threshold access: PSIW was stated to be easily accessible. Referrals were reported to be predominantly performed by the company doctor. Due to the missing sterile hospital atmosphere, service users can be more relaxed (Quote C3.1–3);Preventive character: The company health promotion team experienced PSIW as a tool to reach individuals early in the course of disease. External psychotherapists reported it as a means to prevent severe mental disorders and chronification (C3.4–5);Convenient and simple handling: local closeness of worksite, rapid appointment allocation, little expenditure of time, and easy integration into daily work routine were stated to contribute to convenient and simple handling of PSIW from all three perspectives (Quote C3.6–8);Confidentiality and sense of security: Users and staff alike described the offer as being confidential, which means that they users can have access to it, without being recognized by colleagues. Having the possibility to go there unseen was named an important feature of the service (Table A3, Quote C3.9–11). Additionally, the participants reported that a feeling of mutual trust is necessary for the functioning of the PSIW. Security and trust (of users) are established by clarifying roles and tasks and offering a reliable and punctual service. (Table A3, Quote C3.12–14);Quick fix: PSIW was considered as a quick and prompt answer to acute mental health problems without getting through a lot of bureaucracy for a first appointment (Quote C3.15, -16, -17). For an overview of quotations on the above-mentioned points, see Appendix B, Table A3.

## 4. Discussion

We found that PSIW appears to be a plain and simple intersectoral offer but must deal with several conflicting interests: the employer must balance between support and control; the employee between obtaining support while keeping privacy and the two involved sectors between competition and collaboration. Regarding these aspects and as stated above, our participants reported a favorable evaluation of the PSIW, but also mentioned potential or even reoccurring problems and conflicts. In the following section we discuss potential problems concerning the main elements that emerged in our study.

### 4.1. Specific Problems of the PSIW and Possible Solutions

As we showed in our results, there are some problems concerning PSIW which need solutions to enhance PSIW. Often, certain aspects of the PSIW can be interpreted both as unique advantages and as disadvantages, depending on the point of view, role, and subjective perception of the stakeholder. Therefore, we would like to outline these fields of tension in the following section, discuss the disadvantages from different perspectives, and try to find solutions.

#### 4.1.1. Problems in the Field of Implementation (A)

The results showed that part of the preparation of PSIW is a prior needs assessment and a sensitization in a way that the organization and stakeholders must open up for the intervention. There often is a tension between an unrealistic and rising need for treatment and the realistic possibilities and capacities of PSIW. It is therefore important to provide detailed and transparent information about what PSIW can perform and who its target group is.

Another tension described in the interviews was between ‘need for privacy vs. closeness to the workplace’. This conflict should be considered when implementing a PSIW. While employees were concerned about being seen on their way to therapy, psychotherapists saw the proximity to the work environment primarily as a strength, in order to gain an insight into the reality of work life of employees. To reconcile these two needs, the PSIW can possibly offer therapy sessions also outside the workplace, for example in the rooms of the provider of the PSIW. Vice versa, one possibility to facilitate treatment by making the therapist familiar with the working environment can be by offering factory tours for the therapists as an opportunity to become familiar with the workplace.

#### 4.1.2. Problems in the Field of Persons Involved and Perceptions (B)

Problems can also occur when the persons involved are perceived in an ambiguous way. This was evident in our data when, for example, the service users’ perception of the role of the company doctor differed too much from the self-perception of the doctor. This for example can lead to problems of trust when employees perceived the doctor as being more loyal to the company than to the employees. Such problems should first of all be clarified by the persons themselves, and if possible also by the company. Issues of trust might also generalize on other providers of the PSIW, when the relation between employee and company are too damaged. If trust cannot be restored, PSOC might be a more effective alternative.

Under point 3.2.5 we stated several general problems which were named in the interviews. Namely these were a limited time frame, organizational challenges, disappointment, limited power to influence the organizational part of work-related problems, and misuse of PSIW. The interviewees addressed these issues directly. These problematic areas should be considered in the further development of the PSIW. Making the possibilities and limitations of PSIW clear to all stakeholders is a prerequisite to avoid unfulfilled expectations and disappointment.

Nevertheless, it should be examined whether psychotherapists can be provided with a framework within the organization where they can discuss frequent work-related problems which are not related to the employees but to the working conditions and work organization.

#### 4.1.3. Problems Concerning Meaning, Function, and Context (C)

Under C1 we analyzed areas of potential conflict. Firstly, a tension between PSIW as enriching vs. competing to the company health program, was described. This means that conflicts between the company health promotion team and the PSIW employee can occur, if responsibilities are not clearly defined. Some points seem necessary here to provide a fruitful collaboration of the PSIW and other actors in the company and prevent such conflicts: 1. The employees of the company health program should be closely involved in the implementation of the PSIW; 2. responsibilities and duties should be mutually clarified between employees of the PSIW and the company. This seems to be essential to ensure that PSIW complements the company’s offerings meaningfully without disrupting existing structures.

Secondly, another problem field identified in the interviews is the role of PSIW as a bridge between company health promotion and PSOC. From a company’s perspective, PSOC was often seen as fear-inducing and difficult to access, especially for patients with ambivalent motivation for treatment. Here again, it is necessary to clarify existing limitation of the PSIW. For reasons of duration of treatment, it cannot replace the regular care of mentally ill people. The limitations of the offer should therefore be openly discussed with company health service and service users to adapt expectations and provide an eventual transfer of a service user to PSOC. Furthermore, structures between PSIW and PSOC should be created to facilitate a pathway to further treatment. Patients who have used PSIW as a low-threshold offer can be motivated and supported in finding therapists in the regular care system.

Finally, the problem of the role of mental illness in the social context exists. Mental illness is strongly stigmatized and misunderstood in many social structures and it often takes time to develop an understanding of the subject. Therefore, a more open perception of mental illness (in a company) is important for the successful establishment and use of PSIW in the company. If service users feel ashamed when visiting a PSIW, they will probably not use it. One option is to run additional campaigns to raise awareness about mental illness at the time of the introduction of PSIW.

### 4.2. Synthesis

Regarding PSIW intervention, the context is crucial for understanding the mechanism of change because it is an intersectoral offer. Even though it is designed for the individual, the offer and its meaningful characteristics largely depend on its context. From an organizational perspective, there is a strong input if and how the offer works. This is why there is a need to describe a conceptual framework for organizational interventions [50]. Furthermore, the work environment can also act as barrier to change [51]. Comparable offers, implemented so far under study conditions, lack this information unlike the authors’ report process evaluation. There is scarce evidence from similar early intervention offers that emerged under routine conditions. Published descriptions usually cover those topics in much shorter sections and mostly under methods. Bode et al. described their setting [34] as well as Burman-Roy, who shortly described an occupational psychiatry clinic within an occupational health department in London [32].

To benefit the most from the intersectoral nature of the PSIW offer, it is crucial to define a framework and discuss the issues we written under A2 Organizational and Structural framework. Anticipating areas of potential conflicts can help to find which representative should be involved. We found that making PSIW part of an existing health promotion program is crucial. This goes hand in hand with the conclusion by Rothe and colleagues [52] which stated that implementing offers not being embedded in a company´s health promotion concept might have the opposite effects, thus causing uncertainty and resistance towards the topic. Another important facilitator or barrier is the attitude towards mental health issues within the company. Mental health literacy of employees does not guarantee utilization [53]. Therefore, implementation of the PSIW offer again needs other components such as trainings, for example, to sensitize leaders, other representatives such as members of the work council, the social service, etc.

Making the intersectoral offer work means understanding the social space. With our list of involved persons, we propose who should be taken into account, as these persons or the organization itself, such as Damschroder et al. [46] posited, these involved persons are “carriers of cultural, organizational, professional, and individual mind-sets, norms, interests, and affiliations”. Along with other studies [49], we found that each person or role has its own agenda even though they are highly motivated to collaborate; daily practice draws a different picture [54,55].

A question yet to be discussed is the applicability of the PSIW for companies of different size and there is little research on this issue. One attempt to close this gap is a recent study, that tried to examine how a service such as PSIW works in small- or medium-sized companies [44]. Theoretically, certain features of the PSIW described in this article (which primarily refers to larger companies) seem relevant here, as for example sufficient space in the company to guarantee for a certain amount of anonymity or the presence of an operational health physician inside the company as a well-known person of trust for employees.

Through our interviews, the following six main characteristics of the PSIW were identified:

1. Low-threshold access: all interviewed parties appreciated and acknowledged the low-threshold access;

2. Preventive character of the offer: Company professionals as well as professionals working in the mental health care system confirmed the preventive character of the offer. These two aspects comply with the early intervention urgently drafted by the OECD [1]. The statements of the interviews, i.e., the qualitative data of this part of the study, are in line with quantitative measured outcomes of the same study [45]. Regarding symptoms, we found that the offer addressed individuals at higher rates at an earlier stage of disease than usual care does. In addition, compared with usual care, a higher proportion of male help-seekers took advantage of this service [24]. Moreover, we demonstrated that PSIW has great potential to address individuals at the first signs of the emergence of stress at work [25];

3. Convenient and simple handling: Users and providers described the offers as being easy to integrate into daily (working) structures of the participants. The PSIW was easy to reach by the participants and thus compatible with regular working days. Thus, another structural barrier of PSOC can be overcome with a workplace-based offer [56];

4–5. Confidentiality and sense of security: Both of these features are crucial for the offer and need to be installed actively and not simply taken for granted. This holds true for any medical treatment, especially for an initial psychotherapeutic contact [16]. In light of the abovementioned areas of potential conflict, issues of confidentiality should be handled with precaution. Since the offer also requires the participation of an external party, it is very important that this one is very reliable. Any action taken by this party must be foreseeable and reliable, for example, when it comes to time scheduling;

6. Quick fix: At last, PSIW has proven to be a prompt answer to acute mental health problems, avoiding a lot of bureaucracy in order to obtain a first appointment. This has been confirmed and complies with other offers managing waiting periods within 2 weeks for a first interview [34]. In this context, people expressed high-level expectations and were satisfied at the end. Thus, the goal seems to be more than achieved. However, those high levels of satisfaction among service users and the other parties involved might be due to the fact that the offer is tailored to the needs. Since we know that work-related offers are only to a small extent more effective than usual care [18] but mostly not much superior [35,57], this finding shows the true potential of PSIW counteracting the current lack of early and quick intervention. In order to understand this mechanism in more detail, we should start considering measures such as presenteeism, productivity, and mental health instead of only looking at the number of sick leave days.

The six characteristics reported above describe desirable features of the offer. It seems that the PSIW offers useful support of employees as long as these features are fulfilled.

### 4.3. Strengths and Limitations

This study is the first study to develop a comprehensive conceptual framework of a new and collaborative mental health care offer in the workplace. The comprehensive design of the offer, its context, and significant relations were noted in detail by interviewing stakeholders from the company, service users, and collaborating specialists from the existing mental health care system. Yet, there remain some limitations to this study. The overall impression of the offer PSIW appears quite positive, even though a large part of the results deals with ‘areas of potential conflict’. The positive appraisal might be due to the fact that only involved stakeholders were interviewed. However, earlier studies of our group with persons involved and not involved drew a similar picture [33,39]. Further, the involvement of only five service users, through consecutive recruitment, increased the risk of positive selection of persons with high satisfaction regarding the offer. However, reviews analyzing comparable offers found a high client satisfaction via numerous studies [14].

As mentioned above, the interviews were conducted between 2012 and 2014. One limitation therefore can be that some reports refer to an earlier state of PSIW. In our region, PSIW has been further implemented. Recent studies [43] have shown for the time from 2016 to 2019, that the offer is highly accepted by users and indeed provided quick help and symptomatic relief for users and generally confirms the qualitative observations made in the present study with statistical findings. All in all, the investigation was conducted in only one region in Germany with a very limited number of companies and persons involved. Younger age groups were not present in the interviews. Even though we do not expect a systematic variation in results from these respondents, based on our knowledge of the subject this also limits our results for older groups of age; however, existing literature shows, that persons treated in the PSIW are (possibly due to socioeconomic and educational reasons) of higher age than people treated in PSOC [24]. All these things considered our results are not generalized to a nationwide or international scale. This can be achieved by further in-depth research on the topics.

Members of the research group were involved in making the collaborative PSIW model and are convinced that it is beneficial. In order to allow critical discussion, a mixed research team according to the level of involvement and educational background (psychology, sociology, medicine, and epidemiology) was built.

### 4.4. Future Directions

As mentioned above, research on PSIW should be continued. Questions regarding the generalization of our results to other regions or companies are yet to be answered. One attempt to examine these questions in more details is the recently started ‘friaa project’ [44], that investigates the applicability of the PSIW for medium and small companies in regions all over Germany.

## 5. Conclusions

What makes the intersectoral offer PSIW work is a well-dosed interplay of structures, persons, and institutions as well as the corresponding context. Areas of potential conflict within the company, between employer and employee, as well as between company health promotion and the existing health care system should be considered in advance. Addressing these issues helps to prevent them from being obstacles to implementation. In addition, it fosters synergies and the build-up of the aforementioned characteristics of ‘Psychotherapeutic Consultation in the Workplace’. Even though PSIW has only been implemented in central and northern European countries, the concept can be adapted to other global areas when paying attention to the abovementioned aspects of implementation and conflict. Here, basic requirements such as the presence of an occupational health service should exist to guarantee its functioning. However, since realization (e.g., assumption of costs) primarily depends on the company and an institution providing the service, it functions are relatively independent of local factors such as the health care system and can be flexibly applied to varying circumstances.

## Figures and Tables

**Figure 1 ijerph-19-14894-f001:**
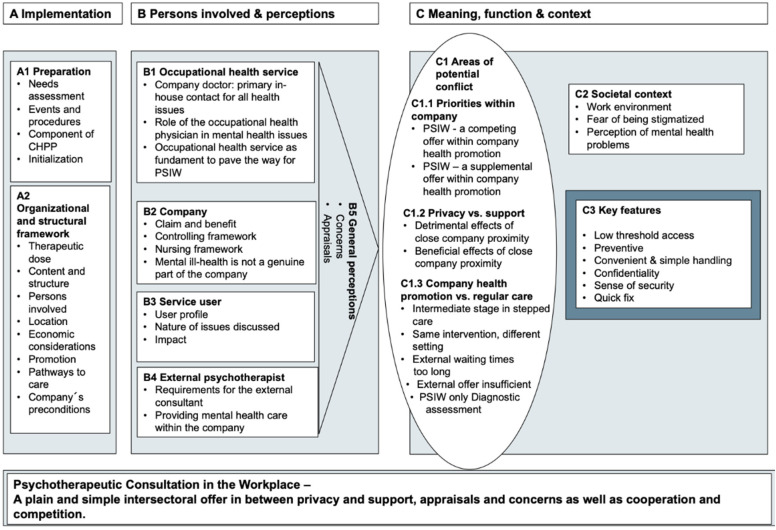
**Conceptual framework of the PSIW.** The conceptual framework comprises the main elements A, B, and C that are important for implementing an intersectoral offer of cooperation between statutory health care and company health promotion. Abbreviations: CHPP—company health promotion program; PSIW—psychotherapeutic consultation in the workplace.

**Table 1 ijerph-19-14894-t001:** Sample characteristics.

Number of Participants: Total Sample N = 20		
**Age**	19–30 years	0
	31–45 years	6
	46–67 years	14
**Sex**	Female	10
	Male	10
**Perspective**	Company health promotion team - Company doctors - Human resource staff - Members of the work council - Other (social service, case management)	3 1 4 1
	Service users: Employees who experienced the offer	5
	External psychotherapist: Medical or psychological psychotherapists	6

## Data Availability

The data presented in this study are available on request from the corresponding author. The data are not publicly available due to participants privacy.

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
