# Peer review of "Conceptual Framework of a Psychotherapeutic Consultation in the Workplace: A Qualitative Study"

_ijerph, 2022, doi:10.3390/ijerph192214894_

Round 1

Reviewer 1 Report

This is a very relevant and well-done study and I appreciate much the development of such frameworks. I have just a minor note for the authors to try to enlarge their conclusions and implications globally with applicable insights to other populations beyond Germany.

Author Response

Dear Reviewer,

thank you for your valuable feedback. Please find our point-to-point answers to your comments below. For a better readability we numbered your comments and occasionally split comments into sub-comments. We integrated all revised parts of our manuscript in this response to allow for an easier reading.  

Please also see the attached file for a comprehensive presentation of our answers. 

Comment 1: This is a very relevant and well-done study and I appreciate much the development of such frameworks. 
I have just a minor note for the authors to try to enlarge their conclusions and implications globally with applicable insights to other populations beyond Germany.
Thank you for your very kind and supportive evaluation of our study. Concerning your comment: We included generalizations beyond Germany in different parts of the text (e.g. Introduction: lines 102-120) and also enlarged the conclusion in the way you proposed. 

It now reads (lines 703-710):

“Even though PSIW has yet been only implemented in central or northern European countries, the concept can be adapted to other global areas when paying attention to the abovementioned aspects of implementation and conflict. Here, basic requirements, such as the presence of an occupational health service should be existing to guarantee its functioning. But since realization (e.g. assumption of costs) primarily depend on the company and an institution providing the service, it functions relatively independent of local factors such as healthcare system and can be flexibly applied to varying circumstances.”

Best regards

Tim Pößnecker

Reviewer 2 Report

The manuscript titled "Conceptual framework of a psychotherapeutic consultation in the workplace: A qualitative study" presents a theoretically and empirically sound study, which is well-written. However, there are some concern points (the majority of them are minor) which I am presenting below.

1. Please check the abbreviation usage. For example, there are some problems with the PSIW (e.g., line 204, 436 and in other places).

2. Please sort the keywords alphabetically.

3. Lines 71-75: please use numbers instead of (i), (ii) ... ; a lack of comma, see line 75: "......into account) (iv)".

4. Table 1. Line with age range of 19-30 is odd. Please, specify the N of the "Company health promotion team" in its each category (i.e., Company doctors, Human resource staff, Members of the work council, Other (social service, case management)).

5. Lines 119-125, please, use the text presented in these lines in the introduction section. This text is not related to the Design and setting section.

6. Section 2.2. Participants and data collection, lines 127-151. Please divide your extremely long paragraph into more paragraphs. Long paragraphs are unwanted. Moreover, provide subsections (participants, location of the study, interviewers, interviewees). In general, structure and specify this section in more detail.

7. Lines 166-167 and in other places. One-sentence paragraphs are undesirable. Please, could you change?

8. Lines 193-198 and see other places: Please could unify the using of "first of all" with secondly and thirdly. Please use "firstly" instead of "first of all".

10. Section 3.3.3. Key features of PSIW (C3): Please describe positive and negative features of PSIW. Please, specify "Confidentiality (Table 4, Quote C3.9, -10, -11) and sense of security (Table 4, Quote 425 C3.14, C3.12, C3.13) were described". It was described, but what does it mean in terms of Confidentiality and sense of security? High levels? Low levels?

11. Describing key features of PSIW, please, try to assess its pros and cons from the perspectives of its availability for different companies. Who can afford these services? Big companies, medium, small ones?

12. Lines 472-502 please, divide this extremely long paragraph. Please, specify and name and number your six main characteristics. "The following six main characteristics were identified". What is it? Main characteristics of what? Please, specify.

13. You have a lot of subsections in the results section, however you have no subsections in the discussion section. It makes your discussion harder to read. Please provide special subsections in the discussion section (i.e., future directions, limitations of the study) and see me other comments on the discussion section below.

14. Please, indicate what solutions you propose to solve some problems of PSIW indicated in the study? Try to name and number these solutions.

15. In general, your manuscript is very very nice and structured. You have a theoretically sound introduction. However, the weakest part of the manuscript is the discussion. In my opinion, it should be more structured and comprehensive. Please indicate specific problems of PSIW and specific solutions, specific positive features and specific negative features from the different perspectives (users, doctors, companies). You have these aspects in the manuscript, but I ask you to specify and structure them.

16. Some theoretical aspects:

1. Please could you describe some analogs of PSIW in different countries?

2. Please could indicate some historical aspects of the PSIW development in Germany?

17. A major concern. You conducted the study from November 2012 until February 2014. The data was obtained 8 years ago. It's been a long time since you conducted the study. Are these data and conclusions up-to-date right now? Please, address this issue carefully in the limitations of the study. Moreover, please, could you report some data on the PSIW development during this time.

18. See "Supplement 1 – Interview guidelines for dealing with therapists/practitioners". What does the "PSIB" mean there?

General discussion.

Dear Authors,

Your manuscript is very very nice and I definitely would like to see it published. I recommend revising the manuscript due to some theoretical and methodological problems.

In general, my congratulations on an excellent submission. You showed mastery of the literature and qualitative analysis and its interpretation. However, I kindly recommend you to address all concerns carefully.

Have a nice day!

Author Response

Dear Reviewer,

thank you for your valuable feedback. Please find our point-to-point answers to your comments below. For a better readability we numbered your comments and occasionally split comments into sub-comments. We integrated all revised parts of our manuscript in this response to allow for an easier reading.  

Please also see the attached file for a comprehensive presentation of our answers. 

Reviewer 2:

Comment 1: Please check the abbreviation usage. For example, there are some problems with the PSIW (e.g., line 204, 436 and in other places).
Thank you for pointing this out. We checked the usage of abbreviations throughout the document and unified it. The only exception from this rule is the end of the conclusion, where we found the unabbreviated form of the word more appropriate.

Comment 2. Please sort the keywords alphabetically.
The keywords have been sorted.

Comment 3. Lines 71-75: please use numbers instead of (i), (ii) ... ; a lack of comma, see line 75: "......into account) (iv)".
The numbers have been changed and now have a better visibility. 

Comment 4. Table 1. Line with age range of 19-30 is odd. Please, specify the N of the "Company health promotion team" in its each category (i.e., Company doctors, Human resource staff, Members of the work council, Other (social service, case management)).
In accordance with a remark of one of the other reviewers we now discussed the missing participants in the age of 19-30 in the methods and limitations (lines 679-685) section. The fact that nobody in the sample fell into this age range might be due to the demographic structure of the sample, which is of older age than regular psychotherapeutic samples. To further illustrate this, we also added some additional literature in this section.

The part in the methods-section now reads (lines 171-174):
“Table 1 gives an overview of the sample characteristics. Possibly due to the age structure of patients treated in the PSIW [24], which is older than in regular PSOC, the age category of 19-30 years was not present in our sample.”

Furthermore, we added the number of the different categories of the company health promotion team. 

Comment 5. Lines 119-125, please, use the text presented in these lines in the introduction section. This text is not related to the Design and setting section.
Thank you for this good observation. We moved the paragraph to the introduction (lines 93-99). 

Comment 6: Section 2.2. Participants and data collection, lines 127-151. Please divide your extremely long paragraph into more paragraphs. Long paragraphs are unwanted. Moreover, provide subsections (participants, location of the study, interviewers, interviewees). In general, structure and specify this section in more detail.

As proposed by your comment, we divided this section into the following subsections: 
2.2. Participants and data collection
2.2.1. Participants and location of study
2.2.2. Interviewers
2.2.3. Data collection
2.2.4. Interview Guideline

We hope that this adjustment fits your demands!

Comment 7. Lines 166-167 and in other places. One-sentence paragraphs are undesirable. Please, could you change?
Thank you for pointing this out. We integrated the respective sentences into the paragraphs above or below.

Comment 8: Lines 193-198 and see other places: Please could unify the using of "first of all" with secondly and thirdly. Please use "firstly" instead of "first of all".
We changed this in the place you mentioned and also in other sections of the manuscript.

Comment 10. Section 3.3.3. Key features of PSIW (C3): Please describe positive and negative features of PSIW. Please, specify "Confidentiality (Table 4, Quote C3.9, -10, -11) and sense of security (Table 4, Quote 425 C3.14, C3.12, C3.13) were described". It was described, but what does it mean in terms of Confidentiality and sense of security? High levels? Low levels?
You are right, that the description of confidentiality and sense of security was rather short. For informing our readers better about the categories discussed, we added some additional explanation. The section now reads (472-478):

“Confidentiality and sense of security: Users and staff alike described the offer as being confidential, which means that they users could have access to it, without being recognized by colleagues. Having the possibility to go there unseen was named an important feature of the service (Table 4, Quote C3.9-11). Additionally, the participants reported that a feeling of mutual trust is necessary for the functioning of the PSIW. Security and trust (of users) are established by clarifying roles and tasks and offering a reliable and punctual service. (Table 4, Quote C3.12-14)”

Comment 11. Describing key features of PSIW, please, try to assess its pros and cons from the perspectives of its availability for different companies. Who can afford these services? Big companies, medium, small ones?
Discussing this further seems a reasonable addition to our discussions section. We therefore added the following paragraph and also included a reference to a more recent study on the topic.
The section now reads as follows (lines 607-614):

“A question yet to be discussed is the applicability of the PSIW for companies of different size and there is yet little research on this issue. One attempt to close this gap is a recent study, that will try to examine how a service like PSIW work in smaller or medium sized companies [54]. Theoretically, certain features of the PSIW described in this article (which primarily refers to larger companies) seem relevant here, as for example sufficient space in the company to guarantee for a certain amount of anonymity or the presence of an operational health physician inside the company as a well-known person of trust for employees.”   

Comment 12. Lines 472-502 please, divide this extremely long paragraph. Please, specify and name and number your six main characteristics. "The following six main characteristics were identified". What is it? Main characteristics of what? Please, specify.
The paragraph you mentioned was indeed very long. In accordance with your remark, we split up the section and added introducing numbers for each main characteristic. Each characteristic now has its own section (lines 616-656).

We also added an explanation in the beginning, clarifying that the main characteristics refer to main characteristics of the PSIW.

Comment 13. You have a lot of subsections in the results section, however you have no subsections in the discussion section. It makes your discussion harder to read. Please provide special subsections in the discussion section (i.e., future directions, limitations of the study) and see me other comments on the discussion section below.
Thank you for giving us such comprehensive feedback on our discussion. Together with your other comments on our discussion, we divided the discussion into the parts you proposed and generally expanded it.

Concerning this comment, we added the following subsections to the discussion:
4. Discussion
4.1. Specific problems of the PSIW and possible solutions
4.1.1. Problems in the field of implementation (A)
4.1.2. Problems in the field of persons involved and perceptions (B)
4.1.3. Problems concerning meaning, function, and context (C)
4.2. Synthesis
4.3. Strengths and limitations
4.4. Future directions

Comment 14. Please, indicate what solutions you propose to solve some problems of PSIW indicated in the study? Try to name and number these solutions.
As mentioned above, we extensively added material to our discussion and re-structured it in many parts. For shedding light on possible problems and solutions of PSIW we added the extensive section “4.1. Specific problems of the PSIW and possible solutions”, where we tried to identify repeatedly mentioned problems mentioned by our participants and discussed possible solutions from our experience of the field.

Since the new section is of long size, we would like you to look at the manuscript (lines 483-575) for all details integrated. 

Comment 15. In general, your manuscript is very very nice and structured. You have a theoretically sound introduction. However, the weakest part of the manuscript is the discussion. In my opinion, it should be more structured and comprehensive. Please indicate specific problems of PSIW and specific solutions, specific positive features and specific negative features from the different perspectives (users, doctors, companies). You have these aspects in the manuscript, but I ask you to specify and structure them.
Thank you for your very kind evaluation of our manuscript! However, we totally agree, that the discussion could be expanded. In accordance with your comment 14 (see line information there), we now discussed our result more extensive. 
We hope, that this broader discussion of our results finds your approval.

Comment 16. Some theoretical aspects:

1. Please could you describe some analogs of PSIW in different countries?
2. Please could indicate some historical aspects of the PSIW development in Germany?
Thank you for your deeper interest int the regional and historical aspect of comparable offers. 

We integrated an additional paragraph in our introduction (lines 102-120) to contextualize the setting of our study regionally and historically. 

The section reads as follows:
“A few ‘PSIW-like’ models have been described in scientific literature for Germany starting with reporting the local history of implementation within a large automobile manufacturer, specific work-related topics/issues at that site [37] and necessities integrated care models at the intersectoral interface like transparency and confidentiality in communication [37,38]. The opportunities and limitations of the services offered have been firstly analyzed in a qualitative analysis by Preiser and colleagues [39]. PSIW was also influenced by evidence from international care offers combining symptom and work-focused interventions [21] describing a model with occupational physicians trained in diagnosis and treatment of mental health problems in the Netherlands. Another Dutch model [20] reported a collaborative care model which combined sickness guidance by occupational physicians with 6-12 sessions of problem-solving treatment, manualized self-help, workplace intervention and antidepressant medication. A work directed and workplace-based model in the USA was reported from Lerner et al [40] focusing on work participation and the reduction of on-work limitation. Interventions strongly involving the workplace with tailored interventions in the return to work process after sick leave due to mental health issues were de-scribed for Denmark [41] and Norway [42]. Besides Preiser [33], none of these studies described in depth the model and its implementation so far even though it is growing like our own model [57] or subject of randomized controlled investigations [42, 54].”

Comment 17. A major concern. You conducted the study from November 2012 until February 2014. The data was obtained 8 years ago. It's been a long time since you conducted the study. Are these data and conclusions up-to-date right now? Please, address this issue carefully in the limitations of the study. Moreover, please, could you report some data on the PSIW development during this time.
As mentioned in our responses to your previous comment, we enriched our discussion by theoretical and empirical aspects. As part of this, we integrated some current studies, showing treatment effects of the PSIW. We reported on this in the in the ‘Strengths and limitations’ (lines 672-677):

“As mentioned above, the interviews were conducted between 2012-2014. One limitation therefore could be, that some reports refer to an earlier state of PSIW. In our region, PSIW has been further implemented today. Recent studies [57] show for the time from 2016-2019, that the offer is still highly accepted by users and indeed provided quick help and symptomatic relief for users and generally confirm the qualitative observations made in the present study with statistical findings.” 

Comment 18. See "Supplement 1 – Interview guidelines for dealing with therapists/practitioners". What does the "PSIB" mean there?
Unfortunately, we used the German abbreviation here, which is PSIB. It has been changed to PSIW.

Best regards

Tim Pößnecker

Reviewer 3 Report

This qualitative study aimed to examine the possibilities for finding a conceptual framework of a psychotherapeutic consultation on the workplace. The authors of this interesting study aimed to establish research "A new healthcare offer called psychotherapeutic consultation in the workplace".  An aim was to find reasons why a study lacks a broader roll-out for example, it might be attributable to undefined context, mechanisms of change, and a lack of communication.

The authors of this study aimed to determine how the new model works and where problems occur.

The present study tried to name the causes that defend successful and effectiveness using psychotherapeutic consultation on the workplace like an early and effective intervention for employees with common mental disorders. The results present that the conceptual framework comprises three main topics: the structured implementation concept; people involved, shaping the concept’s processes; and the meaning and function of the offer within the given context. From the possible conflict themes identified through the analysis, intra-corporate conflicts were identified intra-corporate conflicts; conflicts between company and employee; conflicts between the company health promotion and the health care system. The participants described their experiences using the collaborative PSIW model and were convinced that it was beneficial.

The present study explores semi-structured interviews on motivation, expectations, and experiences with 20 experts involved (members of the company’s health promotion team, service users, and cooperating mental health specialists). I do not consider the sample size to be sufficient for the representativeness to generalise the results for all because the investigation was conducted in only one region in Germany with a very limited number of companies and persons involved.  Other companies or parts of countries or other countries may have different experiences with this issue.

Why did not the 19-30 age group participate in the study?

Is a sample of 20 participants sufficient? Was it not possible to reach more respondents?

The measurements and instruments used by the authors seem to be valid. The processes of analysis and data collection were synchronized. Analysis was performed with atlas.ti. The results are processed in the open-coding process.

The discussion is to some extent and includes the essential findings of the study.  The authors arguments in the discussion are supported by much literature where the authors´ confronted the study s results with the knowledge from the literature.

In view of the limitations of the results reported by the authors, I agree with the authors that it is not possible to generalize the results of the study, as a specific group of respondents participated in it. All in all, the investigation was conducted in only one region in Germany with a very limited number of companies and persons involved.

Do the authors plan further research in the other parts of Germany, other companies or a mixed research team according to the level of involvement and educational background to gain in more detail information that could use for formulation of general conclusions?

The paper I evaluate positively because the presented qualitative study defines structures, to address the needs of the involved persons.  It warns to areas of potential conflict within the company, between employer and employee, as well as between company health promotion and the existing health care system, who the play role whether the implementation of the intervention succeeds. In addition, it fosters synergies and the accumulation of the characteristics of ‘Psychotherapeutic consultation at work’.

Author Response

Dear Reviewer,

thank you for your valuable feedback. Please find our point-to-point answers to your comments below. For a better readability we numbered your comments and occasionally split comments into sub-comments. We integrated all revised parts of our manuscript in this response to allow for an easier reading.  

Please also see the attached file for a comprehensive presentation of our answers. 

Reviewer 3: 

Comment 1: Why did not the 19-30 age group participate in the study?
The absence of this age group from our sample might be attributable to the overall composition of people present in the PSIW, which is – due to social and educational reasons – of older age than in other therapeutic settings. Therefore, the absence of this group might in part be of systematic reason. We added a section on these findings in our ‘Participants and data collection’ section (lines 171-174):

“Table 1 gives an overview of the sample characteristics. Possibly due to the age structure of patients treated in the PSIW [24] and professionals working there, which are both of older than in regular PSOC, the age category of 19-30 years was not present in our sample.”

Nevertheless, this limits our results in a certain way, so we gladly followed your remark and also added a paragraph on this in our limitations (lines 679-685):

“Also, younger age groups were not present in the interviews. Even though we do not expect a systematic variation of results from these respondents, based on our knowledge of the subject, this also limits out results to older groups of age. However, existing literature shows, that persons treated in the PSIW are (possibly due to socioeconomic and educational reasons) of higher age than people treated in PSOC [24].”

Comment 2: Is a sample of 20 participants sufficient? Was it not possible to reach more respondents?
Thank you for your remark! Following Grounded Theory methodology, we finished our data collection after reaching theoretical saturation, which was established after the 20 interviews of the study. Since we did not address this in the first version of the manuscript, we now added a paragraph in the ‘Materials and methods’ section, explaining this more detailed. It reads as follows (lines 191-194):

“The goal was to conduct 20 semi-structured interviews (see Table 1). While conducting the interviews it seemed as if theoretical saturation was reached after conducting the planed number of interviews, since no additional relevant categories emerged from our data anymore.”

Comment 3: Do the authors plan further research in the other parts of Germany, other companies or a mixed research team according to the level of involvement and educational background to gain in more detail information that could use for formulation of general conclusions?
Thank you for your ongoing interests in our research on the subject. We indeed try to deepen our knowledge of the subject and are therefore conducting successive related studies, also in other parts of Germany. To inform our readers about these projects, we added a paragraph in our newly developed ‘Future directions’ section, which is now the final part of our discussion. This paragraph now reads (lines 690-694):

“As mentioned above research on PSIW should be continued. Questions as the generalization of our results to other regions or companies are yet to be answered. One attempt to examine these questions in more details is the recently started ‘friaa project‘ [54], that investigates the applicability of the PSIW for medium and small companies in regions all over Germany.”

We furthermore added another more recent publication of the PSIW to our manuscript, demonstrating the ongoing research on the subject (lines 674-677):

“Recent studies [57] show for the time from 2016-2019, that the offer is still highly accepted by users and indeed provided quick help and symptomatic relief for users and generally confirm the qualitative observations made in the present study with statistical findings”.

Best regards,

Tim Pößnecker
